# CSF Ubiquitin Levels Are Higher in Alzheimer’s Disease than in Frontotemporal Dementia and Reflect the Molecular Subtype in Prion Disease

**DOI:** 10.3390/biom10040497

**Published:** 2020-03-25

**Authors:** Samir Abu-Rumeileh, Patrick Oeckl, Simone Baiardi, Steffen Halbgebauer, Petra Steinacker, Sabina Capellari, Markus Otto, Piero Parchi

**Affiliations:** 1Department of Biomedical and NeuroMotor Sciences (DIBINEM), University of Bologna, 40138 Bologna, Italy; samir.aburumeileh@gmail.com (S.A.-R.); simone.baiardi6@unibo.it (S.B.); sabina.capellari@unibo.it (S.C.); 2Department of Neurology, Ulm University Hospital, 89081 Ulm, Germany; patrick.oeckl@uni-ulm.de (P.O.); steffen.halbgebauer@uni-ulm.de (S.H.); petra.steinacker@uni-ulm.de (P.S.); 3IRCCS Istituto delle Scienze Neurologiche di Bologna, 40139 Bologna, Italy; 4Department of Experimental Diagnostic and Specialty Medicine (DIMES), University of Bologna, 40138 Bologna, Italy

**Keywords:** biomarkers, human prion disease, Creutzfeldt–Jakob disease, Alzheimer’s disease, frontotemporal dementia, neurofilament ligh chain, ubiquitin, mass spectrometry, chitinase-3-like protein 1

## Abstract

Disturbances in the ubiquitin-proteasome system seem to play a role in neurodegenerative dementias (NDs). Previous studies documented an increase of cerebrospinal fluid (CSF) free monoubiquitin in Alzheimer’s disease (AD) and Creutzfeldt–Jakob disease (CJD). However, to date, no study explored this biomarker across the heterogeneous spectrum of prion disease. Using a liquid chromatography−multiple reaction monitoring mass spectrometry, we investigated CSF free monoubiquitin in controls (*n* = 28) and in cases with prion disease (*n* = 84), AD (*n* = 38), and frontotemporal dementia (FTD) (*n* = 30). Furthermore, in CJD subtypes, we evaluated by immunohistochemistry (IHC) the relative extent of brain ubiquitin deposits. Prion disease and, to a lesser extent, AD subjects showed increased levels of CSF free monoubiquitin, whereas FTD cases had median protein values similar to controls. The biomarker showed a good to optimal accuracy in the differential diagnosis between NDs and, most interestingly, between AD and FTD. After stratification, according to molecular subtypes, sporadic CJD VV2 demonstrated significantly higher levels of CSF ubiquitin and more numerous brain ubiquitin deposits at IHC in comparison to the typical and most prevalent MM(V)1 subtype. Moreover, CSF ubiquitin correlated with biomarkers of neurodegeneration and astrogliosis in NDs, and was associated with disease stage but not with survival in prion disease. The differential increase of CSF free monoubiquitin in prion disease subtypes and AD may reflect common, though disease and time-specific, phenomena related to neurodegeneration, such as neuritic damage, dysfunctional proteostasis, and neuroinflammation.

## 1. Introduction

Prion disease, Alzheimer’s disease (AD) and frontotemporal dementia (FTD) spectrum belong to the neurodegenerative dementias (NDs), which are protein misfolding disorders characterized by tissue deposition and intracerebral spread of amyloidogenic protein aggregates [1].

Ubiquitin protein plays a central role in the degradation of proteins in the so-called ubiquitin-proteasome system and is a ubiquitous component of protein amyloid aggregates [2,3,4,5,6]. In physiological conditions, there is a tight regulation of the level of intracellular ubiquitin through the balance between free mono- and conjugated ubiquitin and the respective polyubiquitin chains [5,7]. However, a wealth of evidence suggests that a dysfunctional proteostasis in NDs related to both aging and the inhibitory effect of misfolded protein aggregates may cause an accumulation of ubiquitin in the brain [5,8,9]. Moreover, ubiquitin can be released by dying cells, thereby increasing its level in the cerebrospinal fluid (CSF) as a consequence of neuronal damage [4,5,10].

We have developed a very selective and precise liquid chromatography−multiple reaction monitoring mass spectrometry (LC−MS/MS) method for the measurement of CSF free monoubiquitin and already reported a significant increase in protein levels in Creutzfeldt–Jakob disease (CJD) and AD, in line with other studies investigating the same or other ubiquitin forms [3,4,5,6,9,11]. However, no study to date investigated whether the significant clinicopathological heterogeneity of the prion disease spectrum may influence CSF ubiquitin levels. Indeed, human prion disease encompasses four major phenotypic entities, namely, Creutzfeldt–Jakob disease (CJD), Gerstmann–Sträussler–Scheinker syndrome (GSS), fatal familial insomnia (FFI), and variably protease-sensitive prionopathy (VPSPr) [12]. Moreover, sporadic CJD (sCJD) includes six major clinicopathological subtypes that are mainly determined by the genotype at the methionine (M)/valine (V) polymorphic codon 129 of the *PRNP* gene and the type (1 or 2) of disease-associated prion protein (PrP^Sc^) accumulating in the brain, and named accordingly as MM(V)1, MM2 cortical (MM2C), MM2 thalamic (MM2T), MV2 kuru-type (MV2K), VV1, and VV2 [12].

In the present study, we aimed to validate our previous results in an independent well-characterized cohort, enriched in cases with a neuropathological, genetic, and/or CSF biomarker-based diagnosis of prion disease, AD or FTD. Moreover, we investigated for the first time, the presence, if any, of different ubiquitin profiles among distinct prion disease subtypes and correlated the ubiquitin levels with those of other biomarkers and with clinical variables, such as disease stage and survival. Finally, we provide preliminary data on the relative extent of ubiquitin deposits in the brain of the most common sCJD subtypes.

## 2. Materials and Methods

### 2.1. Case Classification

We retrospectively analyzed CSF samples of 28 healthy controls, 84 subjects with prion disease, 38 with AD, and 30 with FTD submitted to the Neuropathology Laboratory (NP-Lab) at the Institute of Neurological Sciences of Bologna (Italy). All cases were parts of previous studies [13,14]. Informed consent was given by study participants or by their next of kin. The study was conducted in accordance with the Declaration of Helsinki, and the protocol was approved by the Ethics Committees of Area Vasta Emilia Centro (approval number AVEC:18025, 113/2018/OSS/AUSLBO) and Ulm University (approval number 20/10).

Subjects with prion disease were classified according to current European diagnostic criteria [15] and neuropathological consensus criteria [16,17]. They comprised 67 definite cases (54 with a neuropathological diagnosis of sCJD, 1 with VPSPr and 12 carrying a pathogenic *PRNP* mutation (E200K, V210I, D178N, P102L)) and 17 probable sCJD cases (all tested positive by CSF prion real-time quaking-induced conversion (RT-QuIC) assay). For the analysis according to the sCJD molecular subtypes, we merged the subjects with definite sCJD (33 MM(V)1, 12 VV2, 6 MV2K, 2 MM2C, 1 VV1) with those with a probable sCJD diagnosis and a high level of certainty for a given subtype. In detail, the classification was determined by the consensus of three consultant neurologists (SAR, SB, and PP) after reviewing typical clinical features, disease duration at death or at last follow-up, the results of codon 129 genotype (MM, MV and VV), CSF biomarkers, and brain magnetic resonance imaging as previously described [14]. However, to exclude a bias related to possible misdiagnosis, we also analyzed the data after the exclusion of probable cases. Length of survival (disease duration), when available, was calculated as the time (in months) from lumbar puncture (LP) to death or last follow-up. Furthermore, in each prion disease case, we calculated the disease stage as the ratio between the time from disease onset to LP and the disease duration [18]. Then, we classified cases into three categories according to whether LP was performed in the first (disease stage <0.33), second (0.33–0.66), or third (>0.66) tertiles [18].

The diagnosis of AD and FTD clinical spectrum (behavioral variant of frontotemporal dementia, primary progressive aphasia, amyotrophic lateral sclerosis with FTD, corticobasal syndrome, and progressive supranuclear palsy) was made according to established clinical criteria [19,20,21,22,23,24]. All AD cases showed a characteristic AD CSF biomarker profile, based on in-house cut-off values (phosphorylated (p)-tau/amyloid-β (Aβ42) ratio > 0.108 and total(t)-tau/Aβ42 ratio > 0.615) [25]. In the FTD group, 13 subjects carried an FTD-related pathogenetic mutation (9 cases with *C9orf72* and 4 cases with *GRN*), while 17 fulfilled the new criteria for probable 4R- tauopathy [26]. In all FTD cases, the in vivo evidence of coexisting AD pathology was gathered using AD core CSF biomarkers (p-tau/Aβ42 ratio > 0.108) [14]. The control group included 28 subjects lacking any clinical or neuroradiologic evidence of central nervous system disease (e.g., tension-type headache, non-inflammatory polyneuropathies, subjective complaints) and having CSF p-tau, t-tau and Aβ42 in the normal range [14].

### 2.2. CSF Biomarker Analyses

CSF samples were obtained by LP at the L3/L4 or L4/L5 level following a standard procedure, centrifuged in case of blood contamination, divided into aliquots, and stored in polypropylene tubes at −80 °C until analysis.

CSF free monoubiquitin analyses were carried out at the Experimental Neurology Laboratory at Ulm University Hospital in all cases as described [4]. In detail, 50 µL CSF was mixed with an internal standard solution containing ^13^C-labeled ubiquitin and 100 µL acetonitrile. After incubation for 2 min at 1400 rpm, samples were centrifuged at 4000 × *g* for 30 min at RT. The supernatant (70 µL) was diluted with 400 µL water and analyzed by LC-MS/MS. Intra- and inter-assay coefficients of variation (CVs) were <15%.

CSF t-tau, p-tau, Aß42, neurofilament light chain protein (NfL) and chitinase-3-like protein 1 (YKL-40) were analyzed at NP-Lab using commercially available enzyme-linked immunosorbent assay (ELISA) kits (INNOTEST htau-Ag, INNOTEST phosphorylated-Tau181, INNOTEST Aβ1–42, Innogenetics/Fujirebio Europe, Ghent, Belgium; IBL, Hamburg, Germany; R&D Systems, Minneapolis, MN, USA) [13]. ELISA-based biomarkers were measured in all the diagnostic groups except for Aβ42 and p-tau in the prion disease group. The mean intra- and inter-assay CVs were ≤5% and <20% respectively for t-tau, p-tau, Aß42, NfL, YKL-40, as reported [13]. PrP^Sc^ seeding activity was detected by RT-QuIC, as previously described [27,28].

### 2.3. Assessment of Ubiquitin Deposits by Immunohistochemistry

Seven μm thick paraffin-embedded block sections of frontal cortex, anterior striatum, and hippocampus (CA1 sector) from sCJD MM(V)1 (*n* = 5), VV2 (*n* = 5), MV2K (*n* = 5) brains lacking any AD- or Lewy body (LB)-related co-pathology and controls lacking significant neurodegenerative pathology (*n* = 2) were stained with hematoxylin-eosin and processed for immunohistochemistry with a monoclonal anti-ubiquitin antibody with proven specificity [29,30] (MAB1510, dilution 1:1000, Chemicon, Tomecula, CA, USA) as described [27,30]. A mean combined score was given to each case as a result of two operators semi-quantitative assessments of ubiquitin immunopositivity (0, no immunoreactivity; 1, mild; 2, moderate; 3, prominent immunoreactivity).

### 2.4. Statistical Analyses

Statistical analysis was performed using IBM SPSS Statistics (IBM, Armonk, NY, USA, version 21) and GraphPad Prism (GraphPad Software, La Jolla, CA, USA, version 7) software. Based on the presence or not of a normal distribution of the values, data were expressed as mean ± standard deviation (SD) or median and interquartile range (IQR). For continuous variables, depending on the data distribution, the Mann–Whitney U test or the t-test were used to test the differences between the two groups, while the Kruskal–Wallis test (followed by Dunn–Bonferroni post hoc test) or the one-way analysis of variance (ANOVA) (followed by Tukey’s post hoc test) was applied for multiple group comparisons (all reported *p*-values have been adjusted for multiple comparisons). The Chi-Square test was adopted for categorical variables. Multivariate linear regression models were used to adjust (for age and sex) the differences in CSF biomarkers between the groups after the transformation of the dependent variable in the natural logarithmic scale. Receiver operating characteristic (ROC) analyses were performed to establish the diagnostic accuracy, sensitivity, and specificity of each biomarker. The optimal cut-off value for biomarkers was chosen using the maximized Youden index. The Youden index for a cut-off is defined by its sensitivity + specificity − 1. Spearman’s correlations were used to test the possible associations between analyzed variables. For analysis of survival in prion disease, CSF ubiquitin concentration was natural log-transformed to fulfill the normal distribution. We performed univariate and multivariate Cox regression analysis to test the association between time to death and tertiles of CSF ubiquitin and, other well-known prognostic factors in prion disease such as age at LP, sex, disease duration at LP and molecular subtype [31,32]. Given that the analysis limited to a single subtype would have suffered from the small sample size, we chose to consider the whole group of sCJD MM(V)1, VV2 or MV2K types (*n* = 59 cases with available data) and then the single MM(V)1 subgroup (*n* = 33 with available data). The results are presented as hazard ratios (HRs) and 95% confidence intervals (95% CIs). Differences were considered statistically significant at *p* < 0.05.

## 3. Results

### 3.1. CSF Ubiquitin Levels Differ Between Neurodegenerative Dementias

Diagnostic groups showed no significant difference in age and sex. As expected, given the frequent subacute onset and rapid clinical progression, the time interval between onset and LP was significantly shorter in subjects with prion disease than in those with AD or FTD (*p* < 0.001 for each comparison). There was no effect of sex on CSF biomarker levels, and no significant correlation between age and biomarker values in each diagnostic group except for ubiquitin in prion disease (r = 0.419, *p* < 0.001). Therefore, age and sex adjustment were applied for ubiquitin comparisons.

Both prion disease and AD cases showed higher levels of CSF ubiquitin compared to controls (*p* < 0.001 and *p* = 0.003, respectively), whereas in the FTD group ubiquitin values were within the normal range (Table 1, Figure 1A). Moreover, ubiquitin levels were higher in prion disease than in AD and FTD cases (*p* < 0.001 for each comparison), with the latter group showing lower values compared to the former (*p* = 0.002) (Table 1, Figure 1A). These findings were confirmed even after age and sex adjustment. Differences regarding CSF t-tau, NfL, and YKL-40 levels among diagnostic groups confirmed those of our previous study [13].

In the prion disease group ubiquitin levels strongly correlated with t-tau (r = 0.804, *p* < 0.001) and to a lesser extent with NfL (r = 0.499, *p* < 0.001) and YKL-40 (r = 0.525, *p* < 0.001). In the AD group there were correlations between ubiquitin and t-tau (r = 0.689, *p* < 0.001), NfL (r = 0.429, *p* = 0.007) and YKL-40 (r = 0.439, *p* = 0.006). In the FTD group ubiquitin levels correlated with levels of t-tau (r = 0.667, *p* < 0.001), and YKL-40 (r = 0.447, *p* = 0.013).

The diagnostic value of ubiquitin in the discrimination between clinical groups is reported in Table 2. In detail, this biomarker demonstrated good to optimal accuracy in the discrimination between prion disease and other diagnostic groups (area under the curve (AUC = 0.848–0.949) and between AD and controls (AUC = 0.925) or FTD (AUC = 0.880). Diagnostic accuracies of other CSF biomarkers in comparison to ubiquitin were illustrated in Figure 2, with ubiquitin generally showing a lower performance compared to t-tau and/or NfL. Detailed statistics of t-tau, NfL, and YKL-40 were previously reported [13,33].

### 3.2. CSF Ubiquitin Levels Vary in Prion Disease According to Molecular Subtypes and Disease Stage

CSF ubiquitin levels varied significantly between prion disease subtypes (Table 3). After stratification according to sCJD most prevalent molecular subtypes (MM(V)1, VV2 and MV2K), the VV2 group showed higher ubiquitin levels compared to the MM(V)1 (*p* = 0.002) and MV2K (*p* < 0.001) groups, with the former demonstrating increased values compared to the latter (*p* = 0.018) (Figure 1B), even after age and sex adjustment or the exclusion of probable cases. The few cases belonging to other rarer prion disease subtypes (sCJD MM2C, VV1, and VPSPr) showed relatively lower ubiquitin levels compared to the VV2 subtype, but still higher than controls. Ubiquitin levels did not differ between sporadic and genetic prion disease cases. Among the latter group, gCJD V210I subjects demonstrated the highest levels, followed by gCJD E200K and GSS, while FFI cases showed values within the normal range (Figure 1C).

Moreover, the VV2 group showed higher levels of NfL (*p* = 0.001) and YKL-40 (*p* = 0.001) than the MM(V)1 group and higher levels of NfL (*p* = 0.016) and t-tau (*p* = 0.001) than the MV2K group [13] At variance, t-tau and NfL levels were similar between MM(V)1 and VV2 or MV2K, respectively [13]. These findings were confirmed even after the exclusion of probable cases.

In MM(V)1 and MV2K subtypes, the correlations between ubiquitin and neurodegenerative markers share similar coefficients (MM(V)1: NfL r = 0.575, *p* < 0.001; t-tau r = 0.691, *p* < 0.001; MV2K: NfL r = 0.539, *p* = 0.038; t-tau r = 0.679, *p* = 0.005), while they were less powerful in the VV2 group (NfL r = 0.158, *p* = 0.531; t-tau r = 0.546, *p* = 0.019). Furthermore, the correlations between ubiquitin and YKL-40 were comparable in the three main subgroups (MM(V)1: YKL-40 r = 0.422, *p* = 0.015; VV2: YKL-40 r = 0.546, *p* = 0.019; MV2K: YKL-40 r = 0.608, *p* = 0.016). 

In the whole prion disease group, we found a moderate positive correlation between ubiquitin levels and disease stage (r = 0.348, *p* = 0.004). Accordingly, there was a trend toward a gradual increase for ubiquitin levels throughout disease stages, with significantly higher values in the third tertile compared to the first (median third stage 189 pg/mL vs. first stage 90.9 ng/mL, *p* = 0.011), and intermediate values in the second tertile (median 134 pg/mL) (Figure 1D). However, these findings were not maintained after stratification according to the molecular subtype, probably due to the small sample size. However, no difference in the distribution of disease stages was detected between MM(V)1, VV2, and MV2K.

Based on univariate Cox regression analyses (59 prion disease cases, 56 dead, 3 censored) age (HR (CI 95%): 1.058 (1.017–1.100), *p* = 0.005), time from onset to LP (HR (CI 95%): 0.868 (0.790–0.952), *p* = 0.003), and molecular subtype (VV2 vs. MM(V)1, HR (CI 95%): 0.327 (0.162–0.658), *p* = 0.002; MV2K vs. MM(V)1, HR (CI 95%): 0.106 (0.040–0.284), *p* = 0.001) were identified as predictors of the mortality hazard ratios in prion disease, whereas ubiquitin concentration was not associated with survival.

### 3.3. Analysis of Prion-Related Ubiquitin Pathology

To search for abnormal ubiquitin deposits specifically related to CJD pathology, we performed ubiquitin immunohistochemistry on brain sections from the cerebral cortex, hippocampus, and striatum showing various degrees of spongiform change and no or only minimal AD- or LB-related changes. All sCJD sections examined showed a dot- or stub-like ubiquitin immunoreactivity (Figure 3). However, the number and, to some extent, the size of the abnormal deposits varied significantly between cases and anatomical regions, being more numerous and occasionally larger in the areas with more significant spongiform change (Figure 3D–F). Consistently with the more widespread and, on average, more florid pathology that is seen in VV2 and MV2K cases in comparison to the MM(V)1 group, the former subtypes showed a significantly higher score of positive immunoreactivity (VV2: median 6.0, IQR 5.3–6.5; n = 5; MV2K: median 6.05, IQR 5.2–6.5; n = 5) than the MM(V)1 group (median 3.75, IQR 3.5–4.8; n = 5) (MM(V)1 vs. VV2 *p* = 0.010; MM(V)1 vs. MV2K *p* = 0.007; MV2K vs. VV2 *p* = 0.981).

## 4. Discussion

The gold standard for the validation of any new biomarker to be used in the clinical setting requires the evaluation of its performance in independent cohorts. In the present study, we analyzed a large cohort of well-characterized cases to add consistency to previous findings from our and other groups [3,4,5,6,9,11] concerning CSF ubiquitin level variations among NDs. Moreover, we showed for the first time that ubiquitin levels in prion disease differ between sCJD molecular subtypes, and correlate with disease stage. Overall, prion disease cases demonstrated the highest ubiquitin values among diagnostic groups, followed by AD, while FTD cases did not differ from controls as previously reported [4,5]. Interestingly, the median values of the cohort groups largely overlapped with those of our previous publication, which underlines the reliability of the applied analytical method [4].

Furthermore, we confirmed the good diagnostic value of CSF ubiquitin in the differential diagnosis between prion disease and other NDs [4], and, most interestingly, between AD and FTD (AUC = 0.880, sensitivity = 86.8%, specificity = 80.0%), although, overall, the ubiquitin performance remained lower than that of classic biomarkers of neurodegeneration such as NfL and t-tau.

By showing that ubiquitin levels varied significantly across distinct molecular subtypes of sCJD, we also provide new insights into the possible mechanisms associated with CSF ubiquitin increase in NDs. Indeed, sCJD VV2 showed the highest protein values among sCJD subtypes, and, most strikingly, an almost two-fold higher protein concentration than that of classic sCJD MM(V)1, the most common and most aggressive subtype. On one side, the strong correlations between established biomarkers of neuronal/axonal damage, such as t-tau and NfL, and ubiquitin in all NDs and prion subtypes (even in sCJD VV2) indicate that the concentration of free monoubiquitin in the CSF reflects the velocity and extent of neuronal damage [4,5]. Accordingly, we found lower ubiquitin levels in FTD and AD in comparison to prion diseases, and lower protein concentrations in the subtypes with a relatively slow progressive course, such as FFI, GSS, sCJD MV2K, and MM2C, than in sCJD MM(V)1, and VV2 subtypes [12,13,27]. On the other hand, however, some pathological, clinical and biochemical features of the VV2 subgroup suggest that other mechanisms might contribute to the increase in CSF ubiquitin levels, at least in prion disease. Interestingly, sCJD cases demonstrated a dot-like pattern of ubiquitin staining especially in the area with the most significant spongiform change that largely overlaps in morphology and relative distribution with what we previously described for p-tau, the latter being also more consistent in sCJD VV2 and MV2K than in sCJD MM(V)1 [27]. Thus, as p-tau levels correlate with the extent of p-tau deposition [27], the presence and extent of these ubiquitin deposits may also contribute to the raise of ubiquitin concentration in the CSF of sCJD cases. However, the much lower values of CSF ubiquitin in MV2K in comparison to VV2 cases indicate that the latter cannot be the only factor determining the ubiquitin concentration in the CSF. Besides the extent of spongiform change, VV2 cases also typically show a more pronounced and widespread PrP^Sc^ deposition compared to MM(V)1 cases [34]. Therefore, we speculate that this variable might also affect proteostasis resulting in higher brain accumulation of ubiquitin and/or a disproportionate rise of CSF ubiquitin compared to t-tau and NfL in the VV2 group. Accordingly, a positive correlation between CSF and brain ubiquitin concentration has already been reported in AD [9]. Moreover, although sCJD MM(V)1 and VV2 groups share overlapping or slightly different CSF t-tau values [present data, 13, 27], the correlations between ubiquitin and t-tau levels were more robust in the former than in the latter group. Furthermore, given that extracellular ubiquitin may play an anti-inflammatory role [10], the up-regulation of CSF ubiquitin in VV2 may be considered an early attempt to balance a more severe neuroinflammatory response in VV2, which is also reflected by a higher degree of brain microglial immunoreactivity and CSF YKL-40 levels [13,35]. Finally, a positive effect of ubiquitin excess of transcription on its CSF increase appears less probable, given that our previous global gene expression analysis in sCJD brains [36] did not report a significant RNA overexpression of the protein in CJD compared to controls (data not shown). On another issue, the correlation between CSF ubiquitin levels and disease stage suggests a time-dependent increase of the protein, while the lack of association with survival limits its role as a prognostic marker in prion disease.

Given all these considerations, the ultimate role of CSF ubiquitin in neurodegeneration remains to be fully elucidated. However, we suggest that the identification of proteins, such as ubiquitin, as new potential CSF biomarkers, may help us to increase the understanding of distinct pathogenetic pathways involved in neurodegeneration, especially in prion disease.

The major strength of our study is the completeness, and comprehensive characterization of the case series analyzed, which comprise virtually all subtypes of prion disease. In contrast, the cross-sectional nature of the study, which did not help in tracking the longitudinal evolution of biomarker values according to disease stage represents a potential limitation. Moreover, despite the very promising results in terms of both diagnostic value and reliability, the main limit for the implementation of CSF ubiquitin analysis in the clinical diagnostic setting is that LC−MS/MS is often used for drug monitoring but is currently not as distributed as the simple ELISA techniques.

In conclusion, in view of a widespread diffusion of the method, based on our study and on previous findings, we validated and provided further evidence for the role of CSF ubiquitin as a rapid and accurate marker in the differential diagnosis of NDs. Moreover, we contributed to the understanding of possible pathophysiological correlates of CSF ubiquitin changes in NDs. Finally, if future cross-sectional studies will confirm the positive correlation between ubiquitin levels and disease stage in prion disease, its evaluation together with that of other fluid biomarkers may be useful in future clinical trials as a surrogate biomarker of prion disease stage.

## Figures and Tables

**Figure 1 biomolecules-10-00497-f001:**
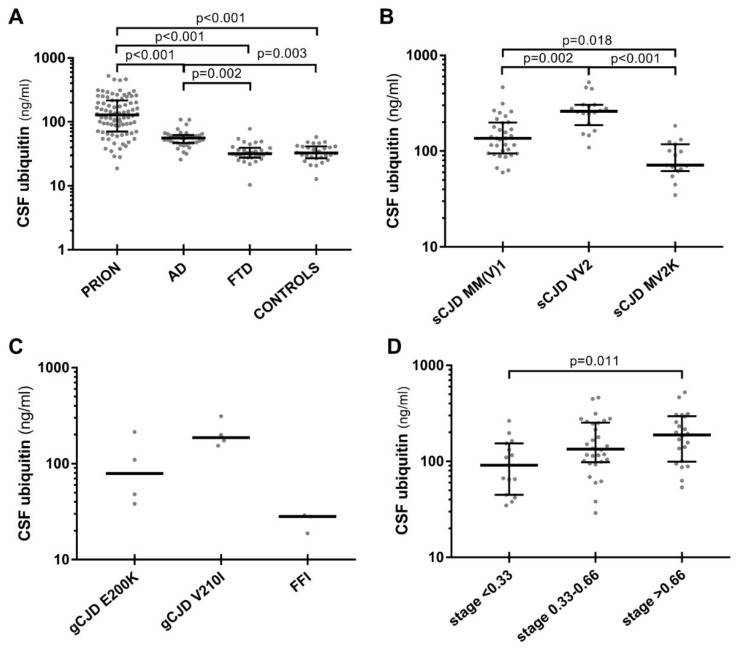
CSF ubiquitin levels in neurodegenerative dementias and in prion disease spectrum. (**A**) CSF ubiquitin levels in prion disease (PRION), Alzheimer’s disease (AD), frontotemporal dementia (FTD) and controls. (**B**) Sporadic Creutzfeldt–Jakob disease (sCJD) VV2 showed significantly higher levels of CSF ubiquitin compared to MM(V)1 and MV2K types. (**C**) CSF ubiquitin in genetic prion disease cases (genetic (g)CJD E200K, gCJD V210I and fatal familial insomnia (FFI)). (**D**) CSF ubiquitin levels increased through disease stages in prion disease. Horizontal lines represent medians. CSF ubiquitin levels are expressed in logarithmic scale. Only statistically significant differences are displayed (Kruskal–Wallis test followed by Dunn–Bonferroni post hoc test).

**Figure 2 biomolecules-10-00497-f002:**
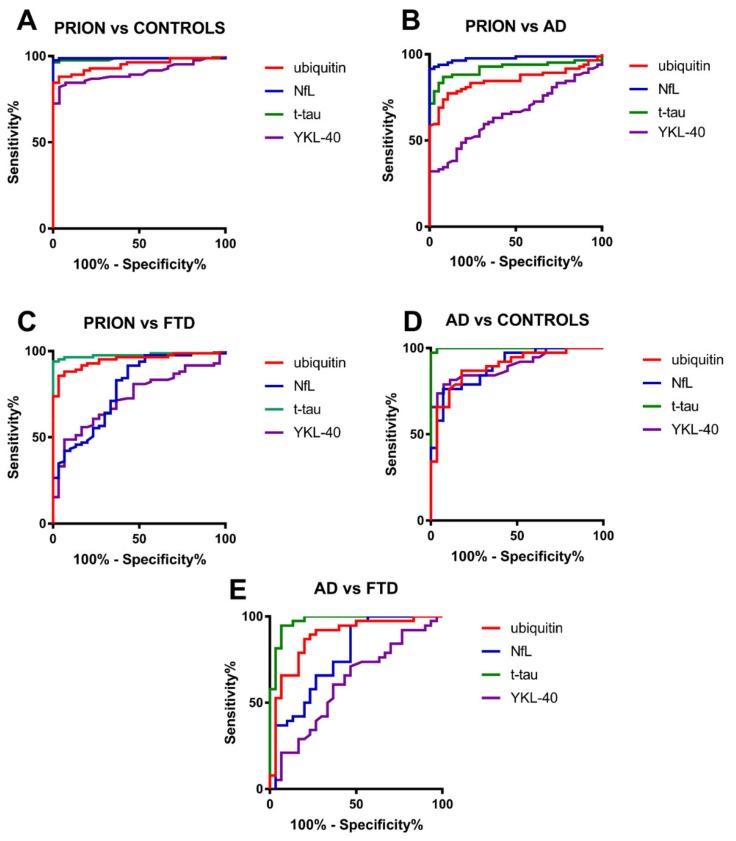
ROC curves of CSF ubiquitin, NfL, t-tau and YKL-40 in the differential diagnosis of neurodegenerative dementias (NDs). (**A**) Comparison between prion disease (PRION) and controls. (**B**) Comparison between PRION and Alzheimer’s disease (AD). (**C**) Comparison between PRION and frontotemporal dementia (FTD). (**D**) Comparison between AD and controls. (**E**) Comparison between AD and FTD.

**Figure 3 biomolecules-10-00497-f003:**
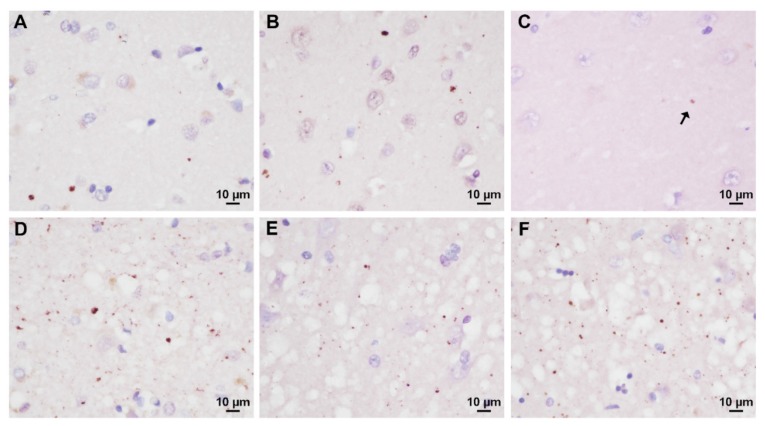
Ubiquitin immunoreactivity in the most common sCJD subtypes. In the neocortex of sCJD VV2, dot-like ubiquitin deposits are sparse and rare in the superficial layers lacking significant spongiform change (**A**), while they are abundant in the deep layers where spongiform change is florid (**D**). Overall, the extent of ubiquitin deposition in the cerebral cortex is lower in sCJD MM(V)1 than in the VV2 subtype (**B**). In the CA1 hippocampal sector, ubiquitin immunoreactivity is almost absent in sCJD MM(V)1 (**C**) the arrow indicates a tiny dot), widespread and prominent in VV2 (**F**), and of intermediate intensity in MV2K (**E**). Remarkably, spongiform change is more pronounced in sCJD MV2K and VV2 (E and F, respectively) than in MM(V)1 cases (C).

**Table 1 biomolecules-10-00497-t001:** Demographic data and cerebrospinal fluid (CSF) biomarker levels in the diagnostic groups.

Diagnosis	Prion Disease	AD	FTD	Controls	P
N	84	38	30	28	
Age at LP (Years ± SD)	67.6 ± 8.95	68.9 ± 7.99	66.5 ± 8.47	64.7 ± 9.89	0.517
Female (%)	46.4%	36.8%	56.7%	46.4%	0.446
Time from Symptom Onset to LP (Months ± SD)	4.54 ± 4.27	47.5 ± 29.37	33.9 ± 22.24	-	<0.001
Ubiquitin Median (IQR) ng/mL	128.5 (70.6–216.3)	55.6 (47.0–62.1)	31.9 (27.7–39.5)	32.6 (27.1–41.3)	<0.001
NfL Median (IQR) pg/mL	6250 (3665–11750)	1199 (856–1574)	2027 (1280–6025)	587 (417–769)	<0.001
t-tau Median (IQR) pg/mL	4573 (1975–9134)	698 (494–1012)	220 (168–288)	165 (138–225)	<0.001
YKL-40 Median (IQR) ng/mL	308 (198–454)	240 (177–289)	186 (150–264)	147 (116–158)	<0.001

**Table 2 biomolecules-10-00497-t002:** Diagnostic accuracy of CSF ubiquitin in the distinction between neurodegenerative dementias.

	AUC		Cut-off (ng/mL)	Sens (%)	Spec (%)
Prion Disease vs. Controls	0.949 ± 0.020	>	51.1	88.1	96.4
AD vs. Controls	0.925 ± 0.032	>	42.5	86.8	88.5
Prion Disease vs. AD	0.848 ± 0.035	>	68.3	77.4	89.5
Prion Disease vs. FTD	0.948 ± 0.020	>	54.6	85.7	96.7
AD vs. FTD	0.880 ± 0.044	>	45.7	86.8	80

**Table 3 biomolecules-10-00497-t003:** CSF biomarkers in prion disease subtypes.

Subtype	N	Ubiquitin (ng/mL) Median (IQR)	NfL (pg/mL) Median (IQR)	t-tau (pg/mL) Median (IQR)	YKL-40 (ng/mL) Median (IQR)
sCJD MM(V)1	33	136.0 (94.6–199.0)	5250 (3689–6800)	6506 (2939–9223)	257 (168–353)
sCJD VV2	18	261.0 (186.3–305.3)	12525 (10750–15963)	8358 (4957–14825)	533 (300–764)
sCJD MV2K	15	71.4 (62.1–118.0)	8250 (3665–11750)	2293 (1433–3088)	336 (199–486)
sCJD MM2C	4	79 (40.7–188.0)	8525 (3112–13763)	2136 (907–6893)	321 (185–976)
sCJD VV1	1	94.7	15900	3790	455
VPSPr	1	65.3	1606	1273	341
gCJD E200K	4	79.0 (40.7–188.0)	8525 (3112–13763)	2137 (907–6893)	321 (185–976)
gCJD V210I	4	186.5 (159.0–283.8)	7813 (2929–12375)	8518 (6069–14125)	409 (165–474)
FFI (D178N)	3	18.8, 29.0, 28.1	11881, 5150, 3536	120, 288, 190	253, 146, 165
GSS (P102L)	1	54.3	2611	5664	450

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
