# Peer review of "CSF Ubiquitin Levels Are Higher in Alzheimer’s Disease than in Frontotemporal Dementia and Reflect the Molecular Subtype in Prion Disease"

_biomolecules, 2020, doi:10.3390/biom10040497_

Round 1
Reviewer 1 Report
This manuscript by Abu-Rumeileh and colleagues validate their previous work exploring CSF ubiquitin levels across several neurodegenerative dementias, and more in depth across different types of prion disease.
The study is well designed and thorough and provides interesting insight into the role of ubiquitin in neurodegenerative dementias. It additionally adds CSF ubiquitin levels as an additional biomarker for several of these diseases.
Author Response
We thank the reviewer for the very positive comments.
Reviewer 2 Report
The manuscript by Abu-Rumeileh et al. is a well-designed analysis of potential biomarkers for the heterogeneous spectrum of CJD. The authors nicely address the importance of specificity, in addition to the sensitivity of a biomarker. This is refreshing!
Generally, I found the methods well explained and the statistical analysis was spot-on! In general, the manuscript is quite good. There are two minor points that should be addressed. First, there are some grammatical and technical writing issues that need to be addressed. For instance, line 53 (However, none the latters) is incorrect English. There was also a couple of instance with the "a" instead of "an" was used in front of a noun that started with a vowel. Minor issues can be fixed with a quick grammar check.
I would recommend staying away from the term "patient" and would refer to the study subjects as "cases". The term "patient" implies the subjects are under clinical care at the time of the study and is more correctly used in case reports. Also, there should always be a space between a number and its units. i.e. 100 mL not 100mL. Lastly, with respect to the IHC, I would recommend running a few positive samples with an isotype control or show that you can compete off the reactivity with the actual protein to confirm the specificity of the monoclonal antibody. This may not be so important in routine pathology examination but for a biomarker paper its important to confirm your antibody is performing properly, given that others may rely on in to utilize the methods described.
Other than that it was difficult to find anything wrong with this manuscript.
Author Response
The manuscript by Abu-Rumeileh et al. is a well-designed analysis of potential biomarkers for the heterogeneous spectrum of CJD. The authors nicely address the importance of specificity, in addition to the sensitivity of a biomarker. This is refreshing!
Generally, I found the methods well explained and the statistical analysis was spot-on! In general, the manuscript is quite good. There are two minor points that should be addressed.
We thank the reviewer for the very positive comments.
First, there are some grammatical and technical writing issues that need to be addressed. For instance, line 53 (However, none the latters) is incorrect English.
We re-checked the manuscript for grammatical errors. Moreover, we revised the syntax of the sentence specifically mentioned by the reviewer (line 53).
There was also a couple of instance with the "a" instead of "an" was used in front of a noun that started with a vowel. Minor issues can be fixed with a quick grammar check.
We performed the grammar check of the whole manuscript, and corrected the grammar error mentioned by the reviewer (lines 104 and 247).
I would recommend staying away from the term "patient" and would refer to the study subjects as "cases". The term "patient" implies the subjects are under clinical care at the time of the study and is more correctly used in case reports.
We substituted the term “patient” with “case” or “subject” throughout the manuscript, as suggested.
Also, there should always be a space between a number and its units. i.e. 100 mL not 100mL.
We inserted a space between numbers according to the reviewer’s suggestion (lines 114-117).
Lastly, with respect to the IHC, I would recommend running a few positive samples with an isotype control or show that you can compete off the reactivity with the actual protein to confirm the specificity of the monoclonal antibody. This may not be so important in routine pathology examination but for a biomarker paper its important to confirm your antibody is performing properly, given that others may rely on in to utilize the methods described.
MAB1510 is a well-known antibody that has been extensively used in the past three decades to study ubiquitin in the post-mortem human brain. The specificity and affinity of this mAb for both conjugated and free ubiquitin have been tested before (see, for example, Morimoto et al., Am J Pathol 1996). Moreover, the antibody and the protocol we used demonstrated the best performance in terms of staining quality in a study by the BNE consortium, in which several antibodies and protocols were compared (see Alafuzoff et al. J Neural Transm 2015). We added these two references to the manuscript.
Other than that it was difficult to find anything wrong with this manuscript.
Reviewer 3 Report
This study explores CSF free monoubiquitin as a biomarker of neurodegenerative dementias , specifically Creutzfeldt-Jakob, Alzheimers and FTD diseases. The authors demonstrated differential increase of CSF Ub in CJD subtypes and AD. Although, the Ub performance is lower than that of classic biomarkers of ND and the method used for CSF ubiquitin analysis, LC-MS/MS, is not always a part of clinical routine, the presented research will definitely elucidate our understanding of possible mechanisms associated with CSF ubiquitin increase in NDs. This is a well written article presenting solid and convincing data.
Author Response

(The authors gave the same response as above.)
